# SGSNet: A Lightweight Depth Completion Network Based on Secondary Guidance and Spatial Fusion

**DOI:** 10.3390/s22176414

**Published:** 2022-08-25

**Authors:** Baifan Chen, Xiaotian Lv, Chongliang Liu, Hao Jiao

**Affiliations:** 1The School of Automation, Central South University, Changsha 410083, China; 2Beijing Institute of Automation Equipment, Beijing 100074, China

**Keywords:** depth completion, secondary guidance, spatial fusion

## Abstract

The depth completion task aims to generate a dense depth map from a sparse depth map and the corresponding RGB image. As a data preprocessing task, obtaining denser depth maps without affecting the real-time performance of downstream tasks is the challenge. In this paper, we propose a lightweight depth completion network based on secondary guidance and spatial fusion named SGSNet. We design the image feature extraction module to better extract features from different scales between and within layers in parallel and to generate guidance features. Then, SGSNet uses the secondary guidance to complete the depth completion. The first guidance uses the lightweight guidance module to quickly guide LiDAR feature extraction with the texture features of RGB images. The second guidance uses the depth information completion module for sparse depth map feature completion and inputs it into the DA-CSPN++ module to complete the dense depth map re-guidance. By using a lightweight bootstrap module, the overall network runs ten times faster than the baseline. The overall network is relatively lightweight, up to thirty frames, which is sufficient to meet the speed needs of large SLAM and three-dimensional reconstruction for sensor data extraction. At the time of submission, the accuracy of the algorithm in SGSNet ranked first in the KITTI ranking of lightweight depth completion methods. It was 37.5% faster than the top published algorithms in the rank and was second in the full ranking.

## 1. Introduction

With the continuous development of 3D computer vision research, the demands for dense depth maps have gradually increased. Therefore, the depth completion task as data preprocessing has received much attention in AR [1,2], VR [3], SLAM [4], and 3D reconstruction [5,6]. Depth completion mainly faces the following three challenges: (1) the current depth completion tasks are all slow and cannot meet the real-time requirements of large projects; (2) RGB image features and LiDAR features are in different modalities. In addition, because they all describe the same scenes, they have a large amount of coupled information, such as relative position relationship and object shapes. These factors all make it hard to fuse these features; (3) The edge blurring problem leads to a large error at the edges of the object in the depth map obtained by the depth completion network.

Researchers have developed a variety of solutions, the most recent of which rely on Convolutional Neural Networks. The original and most commonly used depth completion network is the one-branch fusion network [5,7,8] shown in Figure 1a, by feeding features obtained by concatenating features from two different modalities at the channel end into the depth learning network. This learning method is less effective in learning but runs fast.

However, joint representation alone will result in missing original features. Thus, some researchers have proposed the two-branch fusion network [9,10,11,12,13,14] using coordinated representations for depth completion, whose structural block diagram is shown in Figure 1b. It uses different networks to train their individual features and fuse them through a series of modules. Two-branch fusion network includes the LiDAR feature dominated network [10], the image feature dominated network [11,12], and the balanced extraction fusion network [14]. The latter takes full advantage of the complex texture information of the image as well as the accurate information of the LiDAR to improve performance and therefore attracts more research interest. The balanced extraction fusion network has specific challenges, including (1) the RGB image and LiDAR are two different modalities and (2) the depth map is very sparse. To meet the challenges of depth completion tasks, sparse invariant convolutions [15,16,17,18], uncertainty exploration [19,20], multi-modality fusion strategies [9,13,14], and image guidance strategies [21,22] have been proposed. In addition to these, some approaches also exploit multi-scale features [23,24], surface normal [12,25], or semantic information [26] to further improve performance. Compared with the one-branch fusion networks, this network can better extract RGB image features as well as LiDAR features, but it is less capable of handling coupled features, has more complexity, and usually runs slowly.

In a complex environment, depth information is very sparse. Additionally, due to the large gradient of object edges, the depth information of object boundaries after depth completion is blurred. These two points make depth completion challenging. DynamicConv [27] reduces edge blurring by adaptively learning the importance of different channel features and generating corresponding weights to better cover, express, and exploit the correlation of samples, but its own high computational complexity cannot handle more complex projects. Tang [21] proposed a guidance module in GuideNet that can use the image guide features obtained from the deep learning network to guide sparse depth features into depth information completion. He converted the single guidance into guidance by two convolutional modules, channel-wise conv and cross-channel conv, which substantially reduce the space complexity of this step of guidance, but in principle there is room to reduce the space complexity. Yan proposed the repetitive guidance module [22] by modifying the GuideNet guidance module and repeating it. However, the pure repetition also led to the same increase in spatial complexity and computation time. His proposed network was far from meeting the real-time requirement.

Researchers have also tried other guidance methods.To make better use of local affinities, spatial propagation networks (SPN) [28] have been proposed for image processing networks. However, this propagation method had high complexity. To improve efficiency, a convolutional spatial propagation network (CSPN) [29] was proposed and used for depth completion projects. These two methods had fixed propagation domains and could not fully utilize local affinities. In order to design better convolutional kernels, CSPN++ [30] and NLSPN [5] were proposed. CSPN++ uses convolutional kernels of different sizes to learn the corresponding weights, while NLSPN learns by adaptive convolutional kernels. The DA-CSPN++ [9] proposed in PENet used a better expansion scheme to expand the neighborhood based on CSPN++ and achieve better results. All of these SPN methods were optimized in detail, but when using multiscale optimization, the results become worse because the sparse depth map features that were downsampled are usually not discernible.

On the basis of GuideNet, we designed a lightweight depth completion network based on secondary guidance and spatial fusion for fast and accurate depth completion, whose structure block diagram is shown in Figure 1c, which mainly included the image guidance feature extraction branch, the sparse–coarse branch, and the spatial propagation guidance branch.

SGSNet proposes a lightweight depth completion network based on secondary guidance and spatial fusion borrowing from GuideNet and DA-CSPN++ networks, using the proposed image feature extraction module and the secondary guidance module to make full use of RGB image information and LiDAR information to obtain a more accurate depth map quickly. This network does not require additional datasets, and the KITTI dataset [15,31] can be used directly for the training of the whole network.

The main contributions of SGSNet can be summarized as:An image feature extraction method, which contains a spatial feature extraction as well as a scale feature extraction, is used. Compared with the traditional image pyramid module, this module focuses more on mining richer multiscale information in the same domain.A secondary guidance module is used to guide the guidance features to the LiDAR features and the input DA-CSPN++ network. It is 10 times faster than that in the baseline GuideNet, allowing the overall network to meet the demand for real-time data preprocessing.

## 2. Materials and Methods

We designed an end-to-end deep learning framework based on secondary guidance and spatial fusion for depth completion, whose network structure is shown in Figure 2. The whole framework consisted of three parts: the image guidance feature extraction branch, the sparse–coarse branch, and the spatial propagation guidance branch. The image feature extraction branch meticulously extracts edge information by using the image feature extraction module to obtain lower noise guidance features in parallel and efficiently compared with the baseline GuideNet. The sparse–coarse branch guides the sparse depth features quickly by using the lightweight guidance module to make them informative. Joint representations are performed by the self-attention module. The decoded features and the dense depth map are fed together into the spatial propagation guidance branch. The DA-CSPN++ technique is used in the spatial propagation guidance branch, and the depth information completion module is used to make it more effective and efficient.

In the following subsection, we introduce the image feature extraction module based on image pyramids and the secondary guidance module.

### 2.1. Image Feature Extraction Module

The feature map of these modules is shown in Figure 2. Previous image multiscale feature extraction methods such as traditional image pyramids mostly used simultaneous long and different domain size convolution kernels to extract image features and obtain the multiscale features of the same pixel by comparing the features within the domain of different sizes of the pixel. These methods ignored the multiscale information within the same domain. Unlike these, in SGSNet, we designed the image feature extraction module based on image pyramid, GoogleNet [32] and Vip-DeepLab [33], which can extract more image features while being lightweight. The image feature extraction module in this paper divided it into two steps: spatial feature extraction and scale feature extraction. Spatial feature extraction focuses more on extracting the complete multiscale information in the same domain. In contrast, scale feature extraction complements the features of the pixel by fusing the features of different feature layers for comparison between different domains. This extraction method allows the network to obtain more complete image information.

#### 2.1.1. Spatial Feature Extraction

According to the image pyramid, the image has different image features at different scales. For feature extraction at different scales, the usual method is to first downsample the image, convolve it to obtain the features at that scale, and then upsample it to make the features at different scales the same size. Such a feature is partially complemented by upsampling, which results in some predicted data with great impact on edge information, and does not make full use of all the information at that scale. Therefore, based on the description of atrous convolution in Vip-DeepLab and the method of convolving images using different convolution kernel sizes in GoogleNet, SGSNet used the atrous heterokernel convolution kernel, which is shown schematically in Figure 3.

For the same perceptual domain of a certain element, the feature extraction of the same perceptual domain at different scales is achieved by using different convolution kernel sizes and adjusting the interval of convolution kernel elements. The multiscale information of any domain can be extracted completely and efficiently using the atrous heterogeneous convolution kernel proposed in SGSNet compared with the traditional image pyramid convolution kernel.

#### 2.1.2. Scale Feature Extraction

Spatial feature extraction only extracts the corresponding features at the level of a single feature map, while scale feature extraction extracts different features due to scale transformation at the overall network level, as shown in Figure 2. It first convolves the feature layers at different scales. Then it convolves with the feature layers of the original scale one by one. Finally, this feature layer is made to be more tightly coupled with the feature layers of different feature scales.

### 2.2. Lightweight Guidance Module

The lightweight guidance module was designed based on the guidance module proposed in GuideNet. Compared to the baseline, it can complete the guidance of image to depth map feature extraction more quickly with the same performance. Its main role is to guide the LiDAR features by depth gradient features. The formula for this module is: (1)$$Li_{n+1}=f\left(Li_{n},g\left(Li_{n},GD_{n}\right)\right),$$
where $Li_{n+1}$ and $Li_{n}$ represent the LiDAR features before and after the guidance; $GD_{n}$ represents the guidance features at the current stage; g() represents a filter, and f represents the channelwise conv module. Due to the overlap of the view transformation, the LiDAR features and the guidance features usually have some noise. The role of g() is to filter out the noise of the surrounding elements in the nucleus and use the surrounding image gradient to obtain the guidance features of the pixel in the feature layer, which is fed into the channelwise conv module to complete the guidance of LiDAR features. The function of g() is to obtain the guiding features of the pixel in the feature layer by filtering out the noise of the surrounding elements and using the gradient of the surrounding image, then sending it to the channelwise conv module to complete the guiding of the LiDAR features. The feature map of this module is shown in Figure 4.

It is assumed that the size of the guidance features as well as the sparse depth features are B×C×H×W, where B is the batch size, C is the number of channels, and H and W are the number of input pixels in height and width. It is easy to calculate the complexity of $O(B\times C^{2}\times M^{2}\times H\times W)$ for the direct dynamic convolution of LiDAR features using the guide features, where M is the guide kernel window size. In practical applications of SLAM and 3D reconstruction, higher resolution cameras are used, resulting in larger H and W. The number of channels C of the guide features and LiDAR features in deep learning networks usually increase proportionally with the proportional decrease in H and W, which makes it necessary to reduce the complexity of this guide module in SGSNet. Channelwise conv and cross-channel conv modules reduce the complexity to $O(B\times M^{2}\times H\times W+B\times C^{2})$ by splitting the guidance process into two steps. However, its dynamic filtering greatly affects its operational complexity, because it needs to dynamically filter each channel in the channelwise conv process, which is the origin of $M^{2}$ in its complexity equation. To solve the above problem, the lightweight guidance module in SGSNet first filtered the input data using the shared filter kernel and then sent it to the channelwise conv, so that the dynamic filtering process was not required in the channelwise conv process. The depth gradient feature used to guide the LiDAR features in SGSNet is shown in Equation (Equation 1). The complexity of the lightweight guidance module in SGSNet was O(B×H×W) when passing the channelwise conv, which was reduced by $1/M^{2}$.

Assuming the memory consumption of direct dynamic convolution, the guidance module mentioned in GuideNet and the lightweight guidance module proposed in SGSNet are $M_{DC}$, $M_{GM}$, and $M_{LG}$. Then:(2)$$\frac{M_{LG}}{M_{DC}}=\frac{B\times C\times H\times W+B\times C\times C}{B\times C^{2}\times M^{2}\times H\times W}=\frac{1}{C\times M^{2}}+\frac{1}{H\times W\times M^{2}}$$
(3)$$\frac{M_{LG}}{M_{GM}}=\frac{B\times C\times H\times W+B\times C\times C}{B\times C\times M^{2}\times H\times W+B\times C\times C}=\frac{H\times W+C}{H\times W\times M^{2}+C}$$

From Equation (Equation 3), we can see that our method reduced the complexity significantly. We detail the experimental verification of the complexity reduction in Section 4.

### 2.3. Depth Information Completion Module

The depth map generated by the deep learning network may change the exact pixel depth value originally obtained from the LiDAR data. In order to make better use of this accurate data, DA-CSPN++ is used in SGSNet to second-guide the depth map generated by the first part. On the basis of DA-CSPN++, one modification was made. For the problem of the depth feature loss caused by cropping the depth map obtained from LiDAR by high-dimensional depth feature guidance depth map, SGSNet used a depth information completion module to complement the depth features, so that the accuracy obtained by DA-CSPN++ was higher when using high-latitude features.

The module designed in SGSNet is shown in Figure 5. First, the sparse depth map used atrous heterokernel convolution to obtain the sparse depth features after downsampling. Then, this module used the lightweight guidance module proposed in this paper to guide the sparse depth map for feature completion with the corresponding guidance features. Next, the complemented features were added to the original features, and then the depth map features were complemented by inverse convolution. Finally, the obtained results were entered into the CSPN++ module. Later experiments prove that this step improved the network accuracy, especially in CSPN++ networks that use multidimensional features.

### 2.4. The Training Loss

We used L2 loss for training, and the equation is
(4)$$L\left(\tilde{D}\right)=\left|\right|\left(\tilde{D}-D_{gt}\right)⊙ℜ\left(D_{gt}>0\right)|{|}^{2}.$$

Here, $\tilde{D}$ is the depth map after completion; $D_{gt}$ is the ground truth; ℜ() is an indicator, and ⊙ is elementwise multiplication. We only regressed pixels for which valid data existed in the ground truth.

The ground truth represents different meanings in different branches. In the image guidance feature extraction branch, ground truth represents the depth scale map obtained by processing the dense depth map given by the dataset. In the other two branches, ground truth represents the dense depth map. In the LiDAR feature guidance branch, the train loss is different, and its expressions are: (5)$$L\left(\tilde{D}\right)=L\left(\tilde{D}\right)+\alpha L\left(\tilde{D_{1}}\right),$$
where $L(\tilde{D_{1}})$ is the train loss of the image guided feature extraction branch, and α is a parameter with an initial value of 0.02 changing to $\left\{0.005,0\right\}$ when the epoch is $\left\{3,5\right\}$. The above data were obtained from experience and testing.

## 3. Results

### 3.1. Experimental Setup

#### 3.1.1. Dataset and Evaluation Metrics

Our models were trained and evaluated on the KITTI dataset, which provides over 93,000 sparse depth maps converted from multiline LiDAR and their corresponding RGB images, with a resolution of 1216 × 352. A sparse depth map has about 5% valid pixels, and a ground truth dense depth map has around 16% valid pixels [15]. The dataset contained 86K frames for training, together with 7K validation frames and 1K test frames. In the validation set, 1K frames were officially selected [15,31].

We adopted four metrics for performance evaluation, which were root mean squared error (RMSE [mm]), mean absolute error (MAE [mm]), root mean squared error of the inverse depth (iRMSE [1/km]), and mean absolute error of the inverse depth (iMAE [1/km]). Additionally, the runtime of inference is reported.

#### 3.1.2. Experimental Environment

The model in SGSNet was based on the Pytorch [34] framework. We used two RTX3090s for training, and the trained model was tested on an RTX2080Ti for speed. We used the ADAM optimizer [35] to assist in the training, with the following parameters: $\beta_{1}=0.9$, $\beta_{2}=0.999$, and weight decay of $10^{-6}$. Our model was trained in three stages. First, we trained the image guidance feature extraction branch, setting the batch size to 16 and the learning rate to 0.001. The output used the depth gradient obtained by processing the ground truth, and we trained 10 epochs. Next, we used the parameters from the first step to train the LiDAR feature guidance branch, setting the learning rate to 0.001 and decreasing it by $\left\{\frac{1}{2},\frac{1}{5},\frac{1}{10}\right\}$ when the epoch was $\left\{5,10,20\right\}$, for a total of 30 epochs. After that, the overall network was trained, and the whole learning rate was set to 0.005. The size of the image was reduced by $\left\{\frac{1}{2},\frac{1}{2},\frac{1}{5},\frac{1}{10},\frac{1}{20},\frac{1}{40}\right\}$, when the epoch was $\left\{5,10,20,30,40,50\right\}$. We cropped the image size to half of the original size, set the batch size to 16, and then took the best model. We obtained the above data through experience as well as testing. In the training process, the original data were randomly flipped for better training.

### 3.2. Ablation Studies

We set up a series of ablation experiments to demonstrate the efficiency of our designed image feature extraction module and bidirectional guidance module, and the overall results are shown in Table 1, which were the average results of the tests on the KITTI depth completion validation set. Among them, the baseline was GuideNet; B1 and B2 are the results after adding the spatial feature extraction module and image feature extraction module; C is the result after changing to the lightweight guidance module; C+D1 to C+D4 are the results of the network without the feature guidance module; C+E1 to C+E4 are the results including the feature guidance module.

#### 3.2.1. The Efficiency of the Image Feature Extraction Module

GuideNet was our baseline. Compared with it, we see from the B1 and B2 results that our proposed method had a large improvement in RMSE and MAE and little change in iRMSE and iMAE.

#### 3.2.2. The Efficiency of the Secondary Guidance Module

Based on B2, we evaluated the lightweight guidance module and the deep information completion module in the secondary guidance module.

First, we evaluated the lightweight guidance module and compared the results of B2 and C. Our lightweight guidance module had less improvement in accuracy but had a ninefold improvement in run speed. This also showed the main reason for the network runtime was the guidance module.

Table 2 shows the GPU consumption results of the three different modules obtained experimentally in SGSNet, using 64-bit floats to store the parameters, where B = 1, C = 128, H = 128, W = 608, and using 3 × 3 convolution kernels for progressive convolution. Compared with the direct dynamic convolution, the memory consumption of the lightweight guidance module in SGSNet was reduced from 42.73 GB to 0.035 GB, a reduction of nearly 1155 times. Compared with the benchmark network GuideNet, the memory consumption was reduced from 0.332 GB to 0.035 GB, a reduction of nearly nine times. Subsequent experiments showed that GuideNet only changed the guidance module to 0.035 GB. In addition, subsequent experiments showed that GuideNet only changed the guidance module to the lightweight guidance module in SGSNet, and the running speed was nearly 10 times faster than using its own guidance module with the same accuracy, which can better meet the lightweight requirements of 3D reconstruction and SLAM project for sensor feature extraction.The main reason for the good performance of the lightweight guidance module in SGSNet was the use of a shared filter kernel to preprocess the input module, which reduced the overall complexity of the lightweight guidance module significantly.

Next, we evaluated the depth information completion module. We trained the networks without the depth information completion module but with the addition of the DA-CSPN++ module at three different scales to be the baseline and trained the networks with the depth information completion module to compare with the baseline at the same scale. The results showed that when the depth information completion module was not included and the feature dimension was 4, the effect decreased compared to the feature dimension of 2. However, because of the completion of the depth features, when using the depth information completion module, the results improved. Particularly, at the feature dimension of 4, the improvement in RMSE and MAE was obvious, the RMSE was reduced from 748.93 to 744.04, and the MAE was reduced from 209.82 to 206.30.

Figure 6 shows the effect of the depth information completion module. Figure 6d,e are the dense depth map without the depth information completion module, and the rates are 2 and 4. Figure 6f,g are the dense depth map when the depth information complement module is added, and the rates are 2 and 4. By comparing the details of the figure edges in the yellow box horizontally and vertically, such as the small protrusion at the backpack and the gap between the human leg and the front wheel of the bicycle, we can see that our depth information completion module had better performance in edge blurring improvement. The dense depth map of human body edges obtained by SGSNet was clearer and more accurate.

### 3.3. Comparison with State-Of-The-Art

Our project was first in the KITTI online list for lightweight algorithms and second in the overall online list at the time of publication. We compared the published high-performance algorithms in the KITTI online list, and the results are shown in Table 3 and Figure 7. Our algorithm had a great improvement over other algorithms in the edge details of objects, resulting in more detailed edges and clearer object contours.

Table 3 ranks the RMSE data according to their size. The table also contains the runtime of the algorithms, where the open source algorithms as well as the algorithms in SGSNet were tested for runtime on an RTX2080TI, and the results are bolded. In addition, we used the times filled in on the KITTI online list for the algorithms that were not open source. Among them, the non-open source but top performing networks were all tested using graphics cards with better performance than the RTX2080Ti, and theoretically the runs they got from the test would be faster than those run on the RTX2080Ti. The running time of the algorithm is the time it takes for the network to complete a depth map. Any difference in runtime is the difference in the algorithm itself. The results show that our algorithm outperformed most of the mainstream algorithms on the market in terms of runtime and results, and that the first algorithm had a similar performance while running at one-tenth of the time. The algorithm in SGSNet meets the running time of other large projects such as SLAM, while the first algorithm had very poor real-time performance and could not meet the time requirements of large tasks.

## 4. Discussion

In this paper, we designed an efficient and parallel lightweight depth completion network based on secondary guidance and spatial fusion. It extracted the features of the image in parallel and efficiently through the spatial feature extraction and scale feature extraction. Then, it guided the sparse depth map through the lightweight module for fast LiDAR feature extraction and generated the dense depth map. Finally, the high-dimensional sparse depth map information was complemented by the depth information completion module and input into DA-CSPN++ for secondary guidance. The overall network was efficient and parallel and performed well on the KITTI list.

## Figures and Tables

**Figure 1 sensors-22-06414-f001:**
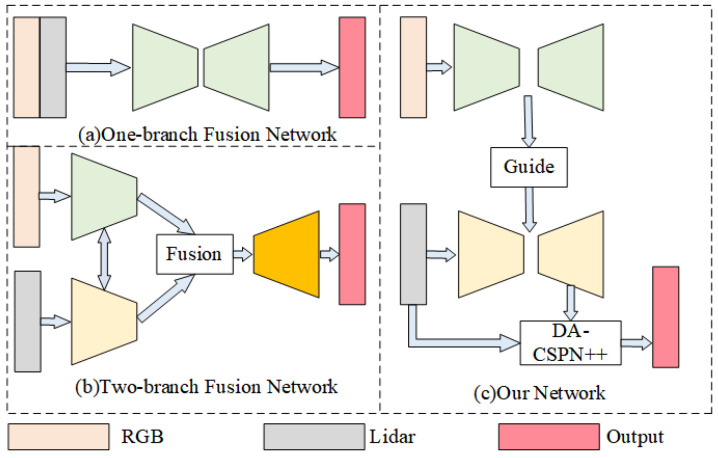
In order to better fuse LiDAR features as well as RGB image features, current depth completion networks mostly use one-branch fusion network (**a**) or two-branch fusion network (**b**). We propose a secondary guidance network (**c**), which aims to fully and efficiently utilize the image as well as LiDAR features to accomplish depth completion more efficiently.

**Figure 2 sensors-22-06414-f002:**
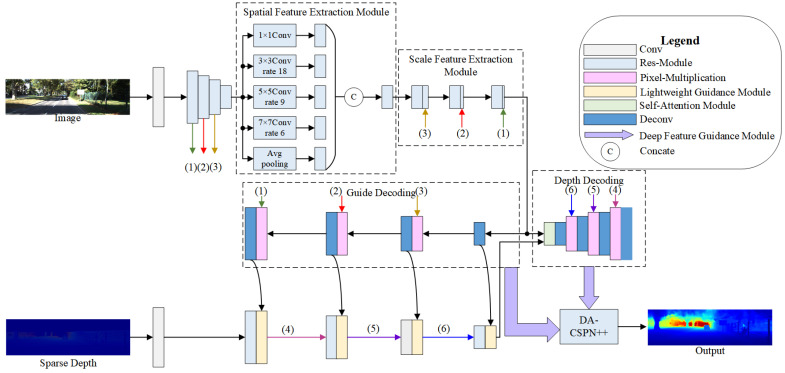
Overview of a lightweight depth completion network based on secondary guidance and spatial fusion. It contains an image guidance feature extraction branch, a LiDAR feature guidance branch, and a spatial propagation guidance branch. It reguides the output depth map by more efficient feature extraction for RGB images, faster guidance for LiDAR, and a depth information completion module. (1)–(6) represent lines.

**Figure 3 sensors-22-06414-f003:**
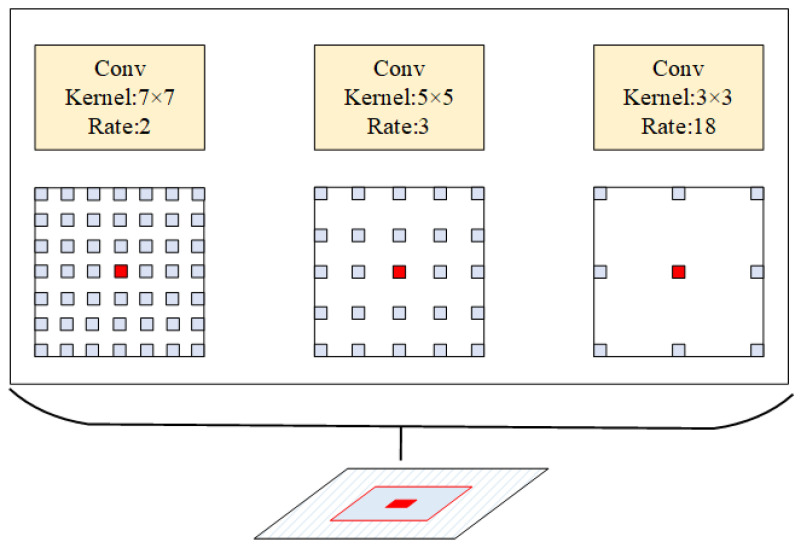
Schematic diagram of the atrous heterokernel convolution kernel. It extracts features under different spaces of the image in the same sensory domain.

**Figure 4 sensors-22-06414-f004:**
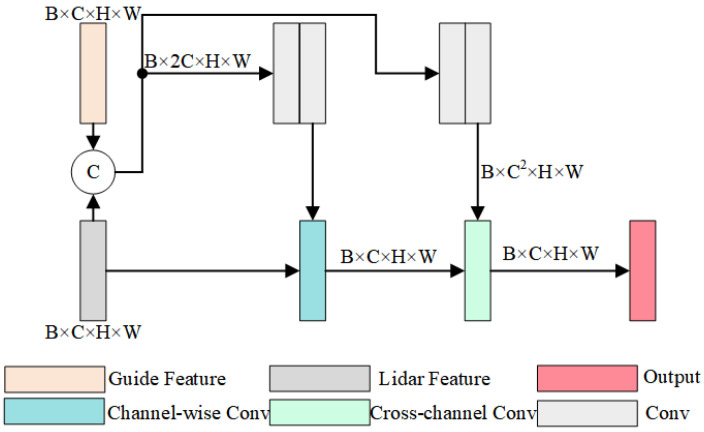
Structural diagram of the lightweight guidance module. It guides features to LiDAR features faster and more efficiently by sharing filtering kernels.

**Figure 5 sensors-22-06414-f005:**
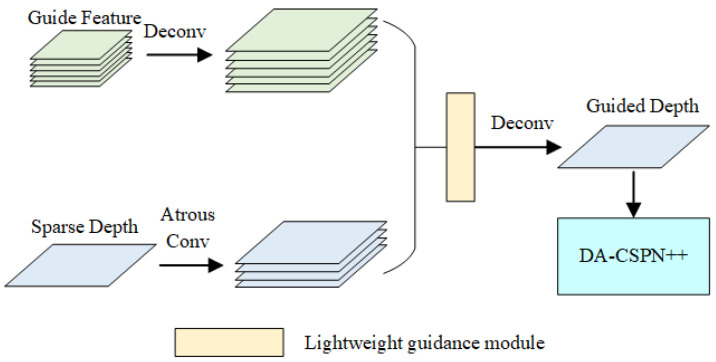
Schematic diagram of the structure of the depth information completion module. The high-dimensional depth map information input to DA-CSPN++ is complemented by the null convolution as well as the same dimensional guidance features to make the high-latitude features have a better positive effect on the redirection of the dense depth map.

**Figure 6 sensors-22-06414-f006:**
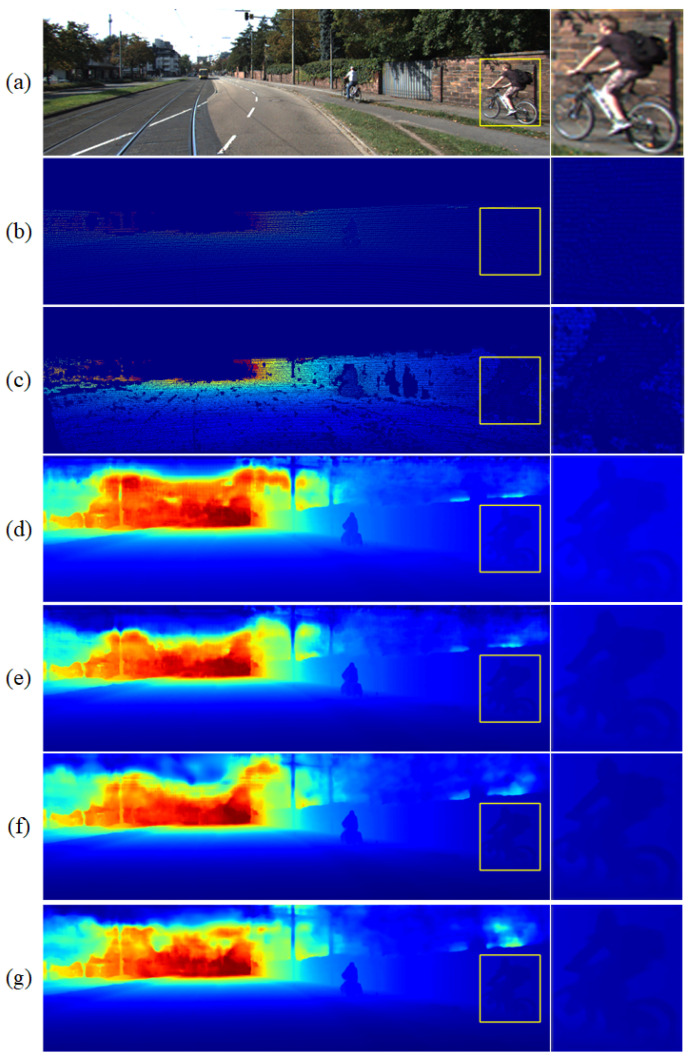
The effect of the depth information completion module where (**a**) is the RGB image, (**b**) is the input sparse depth map, and (**c**) is the ground truth; (**d**,**e**) are the dense depth map without the depth information completion module, and the rates are 2 and 4; (**f**,**g**) are the dense depth map when the depth information complement module is added, and the rates are 2 and 4.

**Figure 7 sensors-22-06414-f007:**
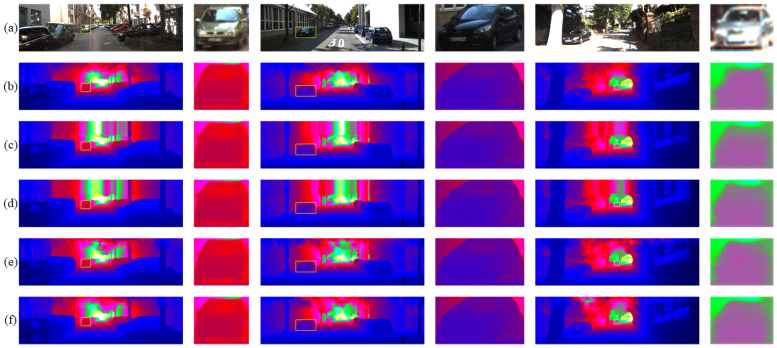
Comparison with the KITTI depth completion test set with state-of-the-art methods, including (**b**) PENet, (**c**) ACMNet, (**d**) GuideNet, and (**e**) CSPN++, where (**a**) is the input RGB image, and (**f**) is the output of our proposed model. As shown by some details, our proposed module has better edge detail features.

**Table 1 sensors-22-06414-t001:** Performance on the KITTI depth completion validation set.

Models	SPFE Module ^1^	SCFE Module ^2^	LG Module ^3^	DIC Module ^4^	DA- CSPN++	RMSE	MAE	iRMSE	iMAE	Runtime
B						778.63	223.37	2.35	0.98	0.140
B1	✓					769.25	220.12	2.34	0.98	0.143
B2	✓	✓				767.04	219.57	2.34	0.98	0.145
C	✓	✓	✓			766.89	218.56	2.31	0.97	0.014
C+D1	✓	✓	✓		1	753.20	210.23	2.21	0.93	0.018
C+D2	✓	✓	✓		1, 2	751.32	209.37	2.17	0.92	0.018
C+D4	✓	✓	✓		1, 2, 4	749.93	209.82	2.18	0.92	0.019
C+E1	✓	✓	✓	✓	1	746.93	209.30	2.18	0.91	0.019
C+E2	✓	✓	✓	✓	1, 2	745.77	208.85	2.16	0.91	0.020
C+E4	✓	✓	✓	✓	1, 2, 4	744.04	206.30	2.16	0.90	0.020

^1^ Spatial feature extraction module. ^2^ Scale feature extraction module. ^3^ Lightweight guidance module. ^4^ Depth information completion module.

**Table 2 sensors-22-06414-t002:** GPU memory consumption for different methods, including dynamic convolution, guidance module, and our lightweight guidance module.

Method	Dynamic Convolution	Guidance Module	Our Module
Memory (GB)	42.73	0.332	0.035
Times	1155	9	1

**Table 3 sensors-22-06414-t003:** Comparisons to state-of-the-art methods on the KITTI test set.

Models	RMSE (mm)	MAE (mm)	iRMSE (1/km)	iMAE (1/km)	Runtime (s)
PwP [25]	777.05	235.17	2.42	1.13	0.100
DSPN [36]	766.74	220.36	2.47	1.03	0.340
**DeepLiDAR** [12]	**758.38**	**226.50**	**2.56**	**1.15**	**0.051**
UberATG [10]	752.88	221.19	2.34	1.14	0.090
**CSPN++** [30]	**743.69**	**209.28**	**2.07**	**0.90**	**0.200**
**NLSPN** [5]	**741.68**	**199.59**	**1.99**	**0.84**	**0.127**
**GuideNet** [21]	**736.24**	**218.83**	**2.25**	**0.99**	**0.140**
FCFR-Net [13]	735.81	217.15	2.20	0.98	0.130
**ACMNet** [14]	**732.99**	**206.80**	**2.08**	**0.90**	**0.330**
**PENet** [9]	**730.08**	**210.55**	**2.17**	**0.94**	**0.032**
RigNet [22]	712.66	203.25	2.08	0.90	0.240
**SGSNet (Ours)**	**723.67**	**209.54**	**2.11**	**0.92**	**0.020**

## Data Availability

Kitti dataset: http://www.cvlibs.net/datasets/kitti, accessed on 15 July 2022.

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
