# Peer review of "SGSNet: A Lightweight Depth Completion Network Based on Secondary Guidance and Spatial Fusion"

_sensors, 2022, doi:10.3390/s22176414_

Round 1

Reviewer 1 Report

Line 64-65 - last line of paragraph is hard to read and the 0.14s is ambiguous as the system creating the time
computation would need more explanation.

Line 78 "are severely missing" - I have never heard those words used together.  Perhaps "are usually not discernible".

Line 93 "in parallel and efficiently".  Efficiently needs to be defined - compared to what?

Line 95-96 "10 times faster than before" - before is ambiguous - at what point in time do you mean?

Line 104 "and efficiently" - compared to what?

Line 132 "and all the data are valid" - should you add this is for the test data and cannot be assumed it would
be true with other data?

line 156 I do not see the definition of B,C,H,W - I know what they are but all readers may not.

line 166 - M2 is nor defined or do you mean M squared

line 199-200 - how was 0.02 determined?

line 224-227 were these numbers obtained via testing, publications, etc.?

line 240 "high requirements on image features" - do you reliance not requirements?

line 255 changeed - spelling

line 264 "to compared with them" - need better wording but I am not sure what

line 269 to 270 - It is an improvement mathematically but is it a statistically valid improvement

Table 3 - no consistency in places past the decimal and significant figures in Runtime(s)

Author Response

Dear Editors and Reviewer1:

Thank you for your letter and for the reviewer1’s comments concerning our manuscript entitled “SGSNet: A Lightweight Depth Completion Network Based on Secondary Guidance and Spatial Fusion” .Those comments are all valuable and very helpful for revising and improving our paper, as well as the important guiding significance to our researches. We have studied comments carefully and have made correction which we hope meet with approval. Revised portion are marked in red in the paper. The major revisions in the paper and responses to reviewers' comments are in the attached word.

Reviewer 2 Report

The paper deals with depth completion networks. The work is based on GuideNet, and the authors claim a modified version which is lightweight and outperforms the original. The work is compared with other results.

First, the paper would improve after additional readings. There are some concerts/questions that appear without references to be later on correctly referenced. To name a few: a) the KITTI dataset appears on line 88, but is referenced on line 204; b) two phrases on lines 293-295 seem that were control one's for the authors and mede it into the paper.

The authors claim two contributions, an image feature extraction method and the secondary guidance module.

* The first one is described in section 2.1, but I cannot infer what is the contribution on the subject. There is only a description of the process which is based on image pyramids. The same thing applies to the lightweight guidance module in section 2.2, which is based on GuideNet but there is no description/analysis of the differences. Moreover, it is not clear the complexity reduction process described on liens 161-166. It does not seem trivial and deserves an explanation. By the way, 1/M2 on line 166 should read M squared.

* The second one is described in section 2.3, and is based on SA-CSPN++. Although these is some description, I can not find in the results any kind of experimental results that support the claims, i.e. those in lines 186-188

In my opinion, the experimental part misses some very important and fundamental questions:

a) section 3.2. Table 1. Do these results correspond to the whole dataset? Only a selected set? On line 239 it is cited "less improvement win runtime" when it is actually a decrease in performance (there is no improvement at all, just the opposite). Table 2. I believe that an explanation is deserved. What is the reason on the better processing/memory performance? The algorithm itself or the implementation on GPU? Figure 7. I can not really appreciate the difference, although when zooming the image. On lines 272-278 there is a claim that a edge blurring improvement occurs, which I can not really distinguish

b) section 3.3. Figure 6. The selected results do appear to be less blurry, but other vehicles in the images do not. I.e. first column, the car on the bottom left appears less blurry on image 'd' than 'f'. I am not sure that these examples can sustain the claims. Table 3. The comparison seems to be inconclusive. Some off the algorithms are run on a RTX2080 while other s are not. First, these should be appropriately differentiated. Second, and in depth analysis of the causes is more than needed: are the differences related to the algorithms or the runtime setup?

Author Response

Dear Editors and Reviewer2:

Thank you for your letter and for the reviewer2’s comments concerning our manuscript entitled “SGSNet: A Lightweight Depth Completion Network Based on Secondary Guidance and Spatial Fusion” .Those comments are all valuable and very helpful for revising and improving our paper, as well as the important guiding significance to our researches. We have studied comments carefully and have made correction which we hope meet with approval. Revised portion are marked in red in the paper. The major revisions in the paper and responses to reviewers' comments are in the attached word.
